# Growth and Competitiveness of ALS-Inhibiting Herbicide-Resistant *Amaranthus retroflexus* L.

**DOI:** 10.3390/plants11192639

**Published:** 2022-10-08

**Authors:** Ruolin Wang, Yujun Han, Ying Sun, Hongjuan Huang, Shouhui Wei, Zhaofeng Huang

**Affiliations:** 1State Key Laboratory for Biology of Plant Diseases and Insect Pests, Institute of Plant Protection, Chinese Academy of Agricultural Sciences, Beijing 100193, China; 2College of Agriculture, Northeast Agricultural University, Harbin 150038, China

**Keywords:** redroot amaranth, ALS-inhibiting herbicide, target-site mutation, resistance, competitiveness, fitness

## Abstract

The evolved acetolactate synthase (ALS) inhibiting herbicide-resistant redroot amaranth has been confirmed in China and caused a great loss in soybean production. This study was conducted to evaluate the growth and competitiveness of ALS-resistant (R) and ALS-susceptible (S) redroot amaranth biotypes. Seeds of both R and S biotypes were subjected to different temperature regimes. Data revealed that the germination percentage and seedling vigor of both biotypes did not differ largely from each other at 10/20 to 30/40 °C. Under noncompetitive conditions, there were no significant leaf number, plant height, or dry weight differences between the R and S biotypes. Moreover, replacement series experiment results indicated that the R and S biotypes have a similar competitive ability. This study shows that there are no significant differences in growth or competitiveness between the R and S redroot amaranth biotypes regarding the physiological characteristics evaluated. Therefore, the proportion and distribution of the R biotype will not be affected in the absence of the ALS-inhibiting herbicide. Some other effective management practices should be adopted to cope with this troublesome weed.

## 1. Introduction

Redroot amaranth (*Amaranthus retroflexus* L.) is a widespread, annual species in the Amaranthaceae family. As a highly competitive C4 species, redroot amaranth can cause significant economic loss to crop plants. It has become one of the most troublesome weeds in soybean fields in northeast China [1]. 

In modern agricultural production, herbicides play a critical role in weed management. The reliance on herbicides for weed control has posed strong selection pressure for herbicide-resistant weeds. There are currently 513 cases of herbicide resistance in different weed species [2]. Acetolactate synthase (ALS) is a critical enzyme that catalyzes the first step of the biosynthesis of the essential amino acids, i.e., valine, leucine, and isoleucine. ALS is also the target enzyme for most commercial herbicides spanning five structurally distinct classes of chemicals. ALS-inhibiting herbicides have been used in various crops and have become popular worldwide due to their high efficacy [3]. In particular, imazethapyr is one of the most frequently used ALS-inhibiting herbicides for weed control in soybean production in China in the last decade. Repeated application of imazethapyr exerted high selection for weed resistance. Unfortunately, redroot amaranth has been identified to be resistant to imazethapyr due to *ALS* site mutation in China in recent years [4]. 

Herbicide resistance traits can affect ecological fitness in resistant biotypes [5], such as the plant growth, competitiveness, and frequency of resistant biotypes [6,7]. A number of studies on the relative competitiveness and fitness between herbicide-resistant and susceptible weed biotypes have been carried out. Many studies have shown that herbicide resistance did not result in fitness cost [8,9,10,11]. However, some herbicide-resistant weed biotypes were reported to be less competitive than susceptible biotypes and may result in growth reduction [12,13,14]. Conversely, Marisa’s study showed that the resistant horseweed (*Conyza canadensis*) biotype might be more vigorous than the susceptible biotype [15]. In such cases, resistant-weed control could be much more difficult. Therefore, it is risky to generalize the relative fitness or competitiveness between herbicide-resistant and susceptible biotypes, and as such, studies are extremely vital for devising appropriate resistance management strategies. 

We have reported four mutations (Ala-205-Val, Trp-574-Leu, Ser-653-Thr, and Asp-376-Glu) in ALS that confer resistance to imazethapyr in different redroot amaranth populations [4,16]. In particular, Trp-574-Leu mutation confers high-level resistance to all the five classes of ALS-inhibiting herbicides [2]. After treating with herbicide, the fitness advantage of resistant redroot amaranth biotype with Trp-574-Leu mutation is obvious, because plants of susceptible biotype will be killed, whereas plants of resistant biotype survive. Unfortunately, no sufficient information about the growth and competitive ability of resistant and susceptible redroot amaranth biotypes is available in the absence of ALS-inhibiting herbicides. Thus, the growth and competitiveness between resistant and susceptible redroot amaranth biotypes were compared in the present study. The main purposes of this study were to: (1) analyze the germination of seeds and seedling vigor of resistant and susceptible biotypes under different temperature regimes; and (2) evaluate competitive ability between the biotypes under competitive conditions. 

## 2. Materials and Methods 

### 2.1. Plant Material

Seeds of the redroot amaranth population that survived imazethapyr treatment were collected from a soybean field in Heilongjiang province of China. The seeds were planted in plastic pots (16 cm diameter) and grown in greenhouses. Seedlings at the 3–4 leaf stage were sprayed with imazethapyr (90 g ai ha^−1^). Twenty surviving plants were individually sampled, genomic DNA was extracted, and PCR was conducted according to Huang et al. [16]. Plants confirmed with heterozygous Trp-574-Leu mutation by *ALS* sequencing were selected and grown for T1 seed production. 

In order to get the genetically uniform ALS-resistant (R) biotype and ALS-susceptible (S) biotypes, T1 plants confirmed with homozygous Trp-574-Leu mutation or no mutation by *ALS* sequencing were isolated to produce seeds for further study. In this way, the purified R and S biotype was obtained.

### 2.2. Germination Assay 

Thirty seeds of each R and S biotypes were placed on filter paper in Petri dishes, while 2 mL water was added. The Petri dishes were kept in five different temperature regimes (10/20, 15/25, 20/30, 25/35, and 30/40 °C) in incubation chambers with a photoperiod of 16/8 h day/night. 

Germination percentage (GP) was expressed as the ratio of the germinated seeds in total tested seeds after germinating for 7 days. The vigor of seedlings was analyzed using the method described by Abdul-Baki [17]. Vigor index (VI) was calculated based on germination percentage (GP), average root length (ARL), and average shoot length (ASL) of the seedlings by the equation: VI = (ARL + ASL) × GP. The experiment was carried out twice with four replications. 

### 2.3. Noncompetitive Study 

Seedlings at the 2 to 3 leaf stage of R and S biotypes were transplanted into plastic pots (16 cm diameter) containing a 1:1 (*V⁄V*) peat: sand sterile. Each pot contained one plant. The pots were placed in a greenhouse at 30 °C day and 25 °C night conditions with a 16 h photoperiod. The plants were watered and fertilized throughout the experiments. 

Three plants (single plant per pot) of each biotype were harvested every 14 days after transplanting, and leaf number and plant height were determined and measured. Above-ground plants were harvested and put in paper bags for drying at 60 °C, and dry weight was measured after drying 2 days. After plants were senesced, seeds were collected from ten individual plants of each biotype and weighed.

### 2.4. Competitive Study

Replacement series experiments were conducted in randomized complete block design with four replications. Seedlings at the 2 to 3 leaf stage of each biotype were transplanted into pots (16 cm diameter, six plants per pot), and seven proportions were chosen: 6R/0S, 5R/1S, 4R/2S, 3R/3S, 2R/4S, 1R/5S, 0R/6S for the biotype mixtures. Plants of R and S biotypes from three pots were collected at 14, 28, 42, 56, and 70 days after transplanting (DAT), respectively, and the number of leaves, plant height, and dry weight were measured. 

This study analyzed the relative crowding coefficient (RCC) [17,18] to determine the competitive ability. Based on the definition, RCC value greater than 1.0 indicates superior competitiveness of R biotype over S biotype; In contrast, when the value is less than 1.0 shows a competitive advantage for the S biotype compared with R biotype. Likewise, an RCC value of 1.0 signifies that the biotypes are relatively equal in competitiveness. The RCC values were calculated according to the formula [19]: 

RCC = (*Y* [5:1R]/*Y* [5:1S] + *Y* [4:2R]/*Y* [4:2S] + *Y* [3:3R]/*Y* [3:3S] + *Y* [2:4R]/*Y* [2:4S] + *Y* [1:5R]/*Y* [1:5S])/(*Y* [6:0R]/*Y* [6:0S]). *Y* indicates the plant’s average height, leaf number, or dry weight. 

### 2.5. Statistical Analysis 

All data obtained from the laboratory and greenhouse were analyzed by ANOVA before further analyses. Data from the two repeated experiments were pooled because no significant difference was observed, and Tukey’s HSD test is used for mean comparison by *p* = 0.05. 

## 3. Results

### 3.1. Germination Experiment

As shown in Table 1, the germination percentage for R and S redroot amaranth biotype was lowest (11–13%) at 10/20 °C, and the highest germination percentage was at 25/35 °C. Both biotypes at the 15/25 and 30/40 °C temperature regime exhibited intermediate germination percentage, and germination percentage exceeded 70% at all tested temperature regimes except 10/20 °C. The germination percentage for S biotype was slightly higher than that of R biotype. However, the difference was not significant. 

Results of seedling vigor, as presented in Table 1, showed that there was no difference between the R and S biotypes. Seedling vigor for both biotypes was highest at 20/30 °C, and lowest at 10/20 °C. The seedling vigor for both biotypes at 20/30 °C was more than 9 times greater than that at 10/20 °C.

### 3.2. Noncompetitive Study 

Leaf number of S biotype was one leaf more than R biotype at the 14 DAT. However, no significant leaf number differences were found on the rest experimental days (Figure 1A). Based on Figure 1B,C, there were no obvious differences in plant height and dry weight between R and S biotypes. Moreover, both plant seed number and seed weight for the R biotype was similar to that for S biotype, and the difference was not significant (Table 2).

Furthermore, the leaf shape of both S and R biotypes ranged from ovate to elliptic and were fairly homogeneous, and no abnormal plant morphological patterns were observed in the R biotype (Figure 2).

### 3.3. Competitive Study

In order to compare the competitive effect between R and S biotypes, replacement series experiments were carried out in this study. As shown in Figure 3A, when R and S biotypes were grown together in different proportions, the leaf number shifted right of the 3:3 ratio, suggesting the S biotype was slightly more competitive than the R biotype. However, replacement series diagrams for plant height and dry weight had the line intersection very close to the 3:3 ratio point, indicating that the S biotype is equally competitive to the R biotype (Figure 3B,C).

Based on Table 3, the RCC values for plant height and dry weight at 14 DAT were slightly higher than 1, revealing that the R biotype was slightly more competitive than the S biotype. Moreover, the RCC values for leaf number, plant height, and dry weight at 28 and 42 DAT were slightly lower than 1, indicating a slight competitive advantage for the S biotype compared with the R biotype (Table 3). In general, the RCC values evaluated at 14, 28, and 42 DAT were approximately equal to 1, indicating the R and S biotypes were relatively equal competitiveness.

## 4. Discussion

The replacement series research is extremely successful to find out the competitive interactions between different biotypes, and is widely used to set a competitive hierarchy in various biotypes [20,21]. In the current study, a replacement series experiment was conducted to compare the competitiveness of R and S redroot amaranth biotypes. When the R and S biotypes were grown together at different ratios, the diagrams for leaf number, plant height, and dry weight had the line intersection very close to the 3:3 ratio point, indicating that the S biotype is equally competitive to R biotype. Previous studies reported that ALS-resistant *C. difformis* was also equally competitive with the susceptible biotype [22]. Moreover, the RCC values in the present study were approximately equal to 1, revealing that the S biotype was likely equally competitive with the R biotype. Based on these replacement series experiments, it could be said that there is no obvious fitness cost for this R biotype and the R and S redroot amaranth biotypes have a similar competitive ability. This equal competitiveness trait between herbicide-resistant and susceptible biotypes is also known for weeds like *C. canadensis* [11], common sunflower [23], and *Conyza canadensis* [24].

Seeds germination of weeds could be stimulated by different temperature regimes [25,26,27]. In the present study, we have tested five range temperature regimes. The optimum germination temperature regime for both R and S biotypes was at 25/35 °C, and it decreases remarkably at the temperature of 10/20 °C. The germination percentage at 25/35 for S biotype (86.6 ± 7.1%) was slightly higher than that of R biotype (81.3 ± 7.7%), which indicated that the S biotype had a slight competitive advantage over the R biotype. In general, both the germination of seeds and vigor of seedlings between R and S biotypes were similar at the five temperature regimes. This finding is in accordance with the research on hairy fleabane that no difference in seed germination between the biotypes [11]. Conversely, the variations in germination were found in herbicide-resistant and susceptible biotypes of weeds. For example, the Kansas R biotype reached 50% and maximum germination 70 and 300 h before Kansas S biotype at 8 °C and 12 and 100 h before the S biotype at 18 °C, respectively [28]. The time to reach 50% germination at 20 °C for three imidazolinone-resistant rice cultivars was 6, 16, and 24 h earlier, respectively, than the susceptible cultivar [29].

No distinguishing characteristics in plant morphology between R and S redroot amaranth biotypes were observed in this study (Figure 2). These results are based on morphology contrast with a previous study reporting the resistant *Amaranthus powellii* plants with Trp-574-Leu mutation have an abnormal leaf morphological pattern due to which they are heterogeneous and smaller as compared to susceptible plants [30]. Throughout the noncompetitive study, there was no significant different plant heights and dry weights between the R and S redroot amaranth biotypes, and similar results have been reported in *C. bonariensis* and other weeds [11,31,32]. Besides, the R redroot amaranth biotype had similar seed production compared to the S biotype, agreeing with previous related studies implying similar production of seeds between the R and S biotypes in weeds like *C. bonariensis* [11] and waterhemp (*Amaranthus rudis*) [33].

In general, the proportion of herbicide-resistant and susceptible weed biotypes could be affected by their growth and seed production. If a resistant biotype has a fitness cost on growth or seed production, the proportion of resistant biotype will be reduced when no selection pressure existed. However, this study on growth and seed production reveals that there is no such fitness cost for R redroot amaranth biotype. Thus, both R and S redroot amaranth biotypes could be present in the field in equal proportions.

The present work showed that the resistance to ALS-inhibiting herbicides did not alter the growth or competitiveness of the R biotype. Therefore, it is possible that the proportion and distribution of the R biotype will not be affected in the absence of the ALS-inhibiting herbicide. It was concluded that ALS-inhibiting herbicide alone cannot control this resistant redroot amaranth efficiently, thus, it should be coped with some other appropriate management practices such as herbicides with different modes of action.

## Figures and Tables

**Figure 1 plants-11-02639-f001:**
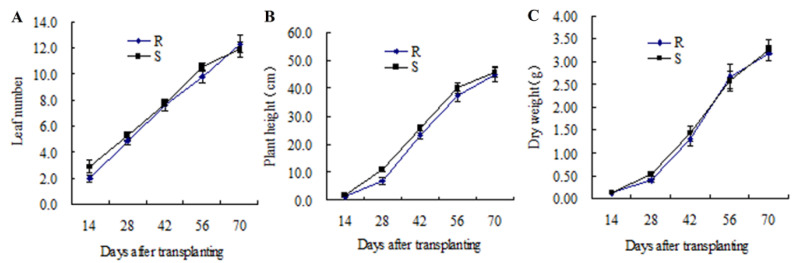
(**A**) Leaf number, (**B**) plant height, (**C**) dry weight of ALS-resistant and ALS-susceptible redroot amaranth biotypes grown under noncompetitive conditions. Bars indicate standard error.

**Figure 2 plants-11-02639-f002:**
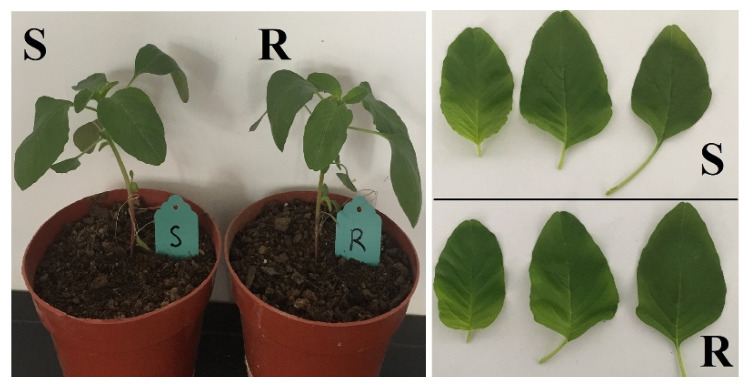
Leaves of ALS-resistant and ALS-susceptible redroot amaranth plants. The sampled leaves were the top three leaves and collected 28 DAT.

**Figure 3 plants-11-02639-f003:**
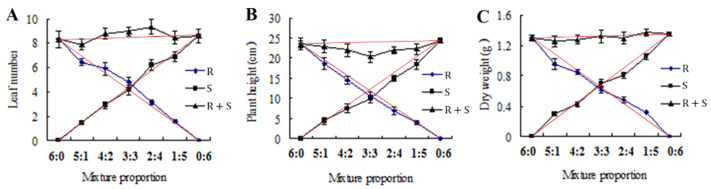
Replacement series diagrams for (**A**) leaf number, (**B**) plant height, and (**C**) dry weight of resistant and susceptible biotypes at 42 DAT. Red lines indicate the hypothetical values for ALS-resistant and ALS-susceptible biotypes with equal competitiveness. Bars represent standard error.

**Table 1 plants-11-02639-t001:** Effect of five different temperature regime on seed germination and seedling vigor (±SE) of ALS-resistant and ALS-susceptible redroot amaranth biotypes.

Temperature Regime (°C)	Germination Percentage (%)		Vigor Index	
	R	S	R	S
10/20	9.3 ± 0.9	9.8 ± 0.7	38 ± 3.9	42 ± 3.2
15/25	72 ± 5.3	77.3 ± 6.1	1369 ± 64.5	1282 ± 78.5
20/30	76.6 ± 6.1	79.3 ± 5.5	3541 ± 178.1	3857 ± 146.3
25/35	81.3 ± 7.7	86.6 ± 7.1	3387 ± 161.9	3345 ± 114.7
30/40	77.6 ± 3.2	81.2 ± 1.2	1796 ± 181.9	1821 ± 109.3

S, susceptible biotype; R, resistant biotype.

**Table 2 plants-11-02639-t002:** Seed production for ALS-resistant and ALS-susceptible biotypes under noncompetitive conditions (±SE).

Biotype	Seed Number/Plant	Seed Weight (g/plant)
R	1389 ± 257 a	0.58 ± 0.02 a
S	1317 ± 206 a	0.57 ± 0.03 a

Means with the same letter in a column indicate that the difference was not significant (*p* = 0.05).

**Table 3 plants-11-02639-t003:** Relative crowding coefficient values for leaf number, plant height, and dry weight at 14, 28, and 42 DAT.

DAT	Relative Crowding Coefficient (±SE)		
	Leaf Number	Plant Height	Dry Weight
14	0.91 ± 0.08	1.19 ± 0.1	1.03 ± 0.06
28	0.95 ± 0.06	0.91 ± 0.05	0.95 ± 0.11
42	0.98 ± 0.13	0.94 ± 0.18	0.96 ± 0.17

## Data Availability

The data are available on request from the corresponding author.

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
