# Peer review of "Growth and Competitiveness of ALS-Inhibiting Herbicide-Resistant Amaranthus retroflexus L."

_plants, 2022, doi:10.3390/plants11192639_

Round 1

Reviewer 1 Report

The problem of the weed in the the agriculture is very important and in particular because they develop resistance against chemical treatment used to preserve the yeld of crops of economic importance. In this commmunication the authors debate on A. retroflexus biotypes R and S. The experiments are well performed, statistical analysis appropriate so as discussion. I think that the communication is suitable for publication in the commemorative special issue.

Author Response

Thank you very much for your appreciation.

Reviewer 2 Report

The authors test the hypothesis that ALS-resistant (R) biotypes of redroot amaranth may exhibit differences in growth or competitiveness relative to wild-type ALS-sensitive (S) biotypes. In a standardized genetic background (via crossing and selection for RR and SS homozygotes), they measured seed germination, seedling vigor, and competitiveness using a replacement series experiment. They conclude that there are no significant differences between the R and S biotypes in the characteristics that they measured, and that the R biotype is therefore unlikely to decrease in the absence of ALS-inhibiting herbicide. 

I found the study to be sound in its science and interpretation, and I have only minor and stylistic comments. 

Seed germination experiment - the authors state that there were no differences in seed germination or vigor index (VI) at any of the temperatures. However, it does look like there is a trend for R to have lower germination %, and possibly lower VI. This makes me wonder if there is a significant overall main effect (or contrast) of biotype across temperatures.  The authors recognize the slightly higher germination % of S in the Discussion. 

Style - In the Discussion, I recommend moving the third paragraph up as the first, and organizing the rest of the Discussion around that paragraph. That would remind the reader of the importance and context of the work, and the subsequent paragraphs could discuss the results and implications of each aspect of the study in more detail (e.g., germination %, morphology, competitiveness, and relationship to other studies). 

Author Response

Seed germination experiment - the authors state that there were no differences in seed germination or vigor index (VI) at any of the temperatures. However, it does look like there is a trend for R to have lower germination %, and possibly lower VI. This makes me wonder if there is a significant overall main effect (or contrast) of biotype across temperatures.  The authors recognize the slightly higher germination % of S in the Discussion. 

Response: Thank you for your comments. We have changed it in the revised manuscript.

Style - In the Discussion, I recommend moving the third paragraph up as the first, and organizing the rest of the Discussion around that paragraph. That would remind the reader of the importance and context of the work, and the subsequent paragraphs could discuss the results and implications of each aspect of the study in more detail (e.g., germination %, morphology, competitiveness, and relationship to other studies). 

Response: Thank you for your comments and suggestion. We have moved the third paragraph up as the first. And add some details in the Discussion.